# Oral Health in migrants children in Melilla, Spain

**DOI:** 10.3390/children10050888

**Published:** 2023-05-16

**Authors:** Gunel Kizi, Ana Raquel Barata, Irene Ventura, Javier Flores-Fraile, David Ribas-Perez, Antonio Castaño-Seiquer

**Affiliations:** 1Egas Moniz, School of Health & Science, Centre for Interdisciplinary Research, 2829-511 Egas Moniz, Portugal; gunelkizi@outlook.com (G.K.); raquelgarciabarata@gmail.com (A.R.B.);; 2Department of Surgery, University of Salamanca, 37008 Salamanca, Spain; 3Department of Stomatology, University of Seville, 41080 Seville, Spain

**Keywords:** Melilla, oral health, migrants

## Abstract

Numerous developing countries’ socioeconomic and political issues resulted in a significant migratory phenomenon, which poses a health burden for the nations that receive migrant populations. Often, the greatest age group of migrants is children and teens. Oral problems are one of the most common reasons that immigrants in the receiving nations visit the healthcare system. Cross-sectional research was conducted on children and teenagers housed at the Temporary Stay Center for Immigrants (CETI) of the Autonomous City of Melilla (Spain) with the aim of identifying the state of the oral cavity of these group of migrants. Information on the condition of the research group’s oral cavity was gathered using the World Health Organization’s standards. The research comprised all of the children and teenagers who were enrolled in the CETI for a defined period of time. A total of 198 children were assessed. It was determined that 86.9% of the youngsters were of Syrian descent. There were 57.6% males and a 7.7 (±4.1) average age. The average caries index for children under the age of six was dft =6.4 (±6.3), and for children aged six to eleven, it was 7.5 (±4.8), taking into account both the temporary and permanent dentition, and for children aged twelve to seventeen, it was 4.7 (±4.0). A total of 50.6% of children between the ages of 6 and 11 needed extractions, compared to 36.8% of children under the age of 6. The population under study had a significant incidence of sextants where bleeding occurred during periodontal probing (mean 3.9 (±2.5)), according to an examination of the community periodontal index (CPI). It is crucial to study the oral cavity status of refugee children when designing intervention programs to improve their oral health and provide health education activities that favour the prevention of oral diseases.

## 1. Introduction

In any health planning process, the situation analysis must be the first step in quantifying the resources needed to carry out an oral health programme [1].

The situation of extreme poverty in which immigrants arrive in the border city of Melilla leads us to assume a precarious state of oral health, but we need objective data to make this fact concrete. Many of these children arrive from populations as far away as Syria, Palestine or Iraq (countries immersed in armed conflicts) and after a difficult journey in subhuman conditions; so, the ultimate objective of carrying out an oral health programme for these children has a marked social character that should be emphasised beyond the scientific value of the study at hand [2].

Healthcare is one of the fundamental priorities of the care provided in refugee camps. War, natural disasters that cause famine or the simple search for a better future for their children cause the civilian population to flee along unsafe routes that lead to physical and psychological decline and a greater propensity to disease. Malnutrition is characteristic of this group [3]. Dental care should not be forgotten in this context in order to improve the deteriorated health of these people [4].

We can differentiate between two types of refugees: those of long duration due to war conflicts that began in previous decades and groups of highly mobile refugees fleeing from different situations as described above [5,6].

An example of projects aimed at “long-term” refugee populations is the so-called Rosengard project [7]. This project is based in Malmo (Sweden) and serves refugee children from conflicts in Somalia, Kosovo, Syria, Iraq, Lebanon, Turkey and Eritrea [8] and an oral health programme was developed for high-risk groups of children.

The high number of refugees arriving in Greece makes it necessary to offer dental solutions to this group. Different institutions such as the College of Dentists of Catalonia and several NGOs created the project of oral care for refugees in Greece (Dentists 4 Refugees) [9]. It has a stable clinic in Khora and also carries out specific actions in the camps. The refugees come from Afghanistan, Syria, Pakistan, Turkey, India, Iraq and Bangladesh.

Currently, the Solidarity Dental Clinic of the Temporary Stay Centre for Migrants (CETI) in Melilla can be considered a solid dental care project for refugees at international level [10].

The Solidarity Dental Clinic of the CETI was created in response to the need for dental care for a population of some 6000–7000 refugees per year in Melilla, one of the southern borders of Europe [10].

Melilla is a Spanish city located in North Africa, bordering the RIF region [11]. It is located within the Tres Forcas peninsula with the province of Nador (Morocco) as the nearest territory. It has a territorial area of 12.5 km^2^ and a population of 86,308 in 2018. Four cultures coexist in Melilla: Christian, Muslim, Jewish and Hindu [11,12,13,14,15].

Due to its geographical location, the city of Melilla is an ultra-peripheral region of the European Union. This makes it one of the gateways to Europe for migrant and refugee populations from Africa and Asia. Migration comes from more than 38 countries. These people either cross the border irregularly or even jump the triple fence that separates Melilla from Morocco [16,17,18,19].

In 1999, the Temporary Stay Centre for Immigrants (CETI) was opened to house all irregular immigrants in Melilla. It is a public administration establishment that provides shelter, services and social benefits to these groups.

The CETI had an initial capacity of 350 people and currently houses 796 immigrants. However, the CETI is normally overcrowded. There were more than 2000 residents in the Centre [20]. Traditionally, refugees came from Algeria, Morocco, Mali, Guinea, Cameroon, Nigeria, Chad, Angola, Gabon, Democratic Republic of Congo, Niger, Zimbabwe, Ivory Coast and from many more sub-Saharan countries and regions [21,22,23]. In recent years, the majority of war refugees came from Syria [24,25]. On average, people temporarily housed in the centre usually spend an average of 3 months in the CETI waiting to be sent to another home in Spain or in the rest of Europe in many cases for family reunification.

All residents are provided with medical, psychological and legal assistance. The resident population receives general culture courses and intensive Spanish courses. There are also various workshops and sports activities. Pre-school children are provided with a day-care service and school-age children attend state-run centres in the city [26]. In the CETI, there is a meal programme run by nutritionists from the Melilla City Government that provides a balanced diet.

The size of the migrant population in the CETI-Melilla and the dental needs it presented led to the joint action of the College of Dentists and Stomatologists of Melilla and the Luis Séiquer Social Dentistry Foundation [10], for the inauguration of the Solidarity Dental Clinic in February 2016, the Solidarity Dental Clinic of the CETI-Melilla was inaugurated [27,28]. This clinic has a pioneering character, serves as an element of social integration and develops interdisciplinary strategies for the promotion of public health [10]. There is a school in the CETI where, since the beginning of the collaboration with the solidarity dental clinic, a joint programme was carried out with the school’s teachers on oral hygiene to promote daily toothbrushing among the children, monitored by their parents.

The only oral health data published so far were those of Goncalves et al. for Syrian immigrant children in 2015, in which the main conclusions of this year already spoke of a high caries prevalence for the Syrian immigrant population, highlighting the need for specific oral health programmes [29]. More recently, in 2018, a study carried out by Sevillano reflected a low oral quality of life for the children housed in this CETI in Melilla [24].

In light of these presumptions, this study was conducted with the objective to determine the oral health of a group of children aged five to thirteen who are residents of the Temporary Stay Center for Immigrants (CETI) in the Autonomous City of Melilla (Spain) and who are at risk of social exclusion because of their status as refugees.

## 2. Materials and Methods

### 2.1. Type of Study and Design

A cross-sectional study was conducted on children and adolescents in refugee conditions who were housed in the CETI of Melilla, Spain. All children and teenagers from one to seventeen years of age who were in the CETI in April and May 2018 were included. No estimation of sample size was made for this study as it was the totality of children accommodated in the CETI that was subjected to descriptive analysis.

The World Health Organisation criteria were used to assess the oral cavity status of the children [30]. Information on dental caries and care needs, periodontal tissue status, dental fluorosis, frequency of brushing and dental visits were included.

Following the WHO recommendations for oral health in its book Oral Health Surveys, Basic Methods, an intra-examiner calibration of the single dentist who carried out the observations was performed, achieving a kappa index of 0.85, suitable for this type of population-based survey [30].

Analyses of dental and periodontal status were carried out by age group, taking into account the correspondence with the type of dentition assumed due to the chronology of eruption. Three groups were formed, one group with presumed primary dentition (<6 years); another group with mixed dentition (6–11 years) and the last group with permanent dentition (12–17 years).

### 2.2. Statistical Methods

For the sample description of quantitative variables, the mean (M) and standard deviation (SD) were used as measures of central tendency and dispersion, respectively. Nominal and ordinal variables were described using the sampling distribution of the count and the percentage it represents.

The variables used were: socio-demographic variables (country of residence, sex, age and type of dentition), clinical variables in relation to caries (caries indices such as dmft or DMFT, restoration index-RI- and Significant index of caries -SiC-), dental clinical variables not associated with caries (periodontal index -CPI-, type of malocclusion and pain) and data associated with oral health such as tooth brushing or visits to the dentist.

Comparison between groups with respect to quantitative variables used Student’s test, while non-parametric tests such as the chi-square test were used to compare groups with respect to qualitative variables. A *p*-value of *p* < 0.05 was set to determine a statistically significant difference and *p* < 0.10 to declare a clear trend towards significance. SPSS v.21 software (Statistical Package for Social Science, Chicago, IL, USA) was used for all analyses.

## 3. Results

A total of 198 children were examined, 57.6% were male. Children aged 1 to 17 years were included. The mean age was 7.7 ± 4.1. Approximately one quarter (25.8%) were in the 6–8 years age group. Table 1 presents the socio-demographic variables of the study group. It was observed that the majority of the children were Syrian (86.9%), followed by the group of Moroccan children (6.1%), children from Palestine, Ivory Coast, Liberia and Algeria were also included, who together accounted for 7% of the study group.

Most of the children studied were in mixed dentition, 44.9%, followed by primary dentition, 34.3% and about one fifth of the children had only permanent teeth present (Table 1).

With regard to the state of the dentition (Table 2), it was observed that in the primary teeth, children under 6 years of age had a significantly higher average number of teeth with caries (6.3 ± 6.3) compared to groups with mixed dentition (4.7 ± 3.9). In contrast, a higher caries index was found in permanent teeth between 12 and 17 years of age (4.4 ± 4.0) than in children with mixed dentition (2.0 ± 2.0).

In all cases, the caries restoration rate was very low (<3%) or even zero. Furthermore, the significant caries index (SiC) showed very high data for the average caries index in primary, mixed and permanent dentition which ranged from 6.3 ± 6.4 observed in children under 6 years of age to 10.1 ± 6.0 recorded in children between 6 and 11 years of age.

According to Table 3, the most common dental treatment needed in the three age groups into which the study group was divided was fillings, but children under the age of 12 also had a high need for extractions (1.8 ± 2.9 in children under the age of 6 and 1.5 ± 2.3 in subjects between 6 and 11 years of age). A total of 50.6% of children between the ages of 6 and 11 needed extractions, compared to 36.8% of children under the age of 6.

A significant portion of young children under the age of six required the filling of three or more teeth (55.9%). In this same group, in relation to endodontics or exodontics, 38.2% needed some invasive treatment (Table 3). It is also important that in participants aged 6–11 years, 75.3% needed three or more fillings and 57.3% required some invasive dental treatment (root canals and/or extractions).

A lower need for dental treatment was recorded in participants aged 12–17 years, more than 80% did not require root canal treatment or extractions; however, 61% were found to have a need for fillings in three or more teeth.

Periodontal status was assessed only in participants aged 12–17 years (Table 4). Bleeding was frequent, more than half of the anterior sextants had bleeding areas, this was higher in the posterior teeth where more than 70% of the indicator teeth in the sextants had bleeding. The presence of calculus was observed in only one participant. Specifically, teeth 16, 26 and 46 showed the most frequent bleeding (73.2%) (Table 4).

Table 5 shows that extraction needs in the primary dentition were significantly higher in boys (2.3 ± 3.3 teeth) than in girls (0.9 ± 2.0). Similarly, the total number of invasive treatments was significantly higher in boys (3.0 ± 4.7) than in girls (1.2 ± 2.4), and the need for space maintainers in the mixed dentition was significantly higher in boys (0.5 ± 1.3 teeth) than in girls (0.1 ± 0.5).

Although the trend is, with respect to periodontal status, that females have a higher number of sextants with bleeding, these differences were not statistically significant (*p* > 0.05) (Table 6).

Table 7 shows the distribution of 12-year-olds according to their attendance at the dentist in the last year by sex. In both male and female groups, the highest percentage was found for children who had never been to the dentist. No statistically significant differences were observed between males and females regarding visits to the dentist (*p* = 0.861).

Table 8 shows that fillings are the main dental treatment needs of the study population, but between genders, these differences were not significant.

Immigrants requiring three or more fillings were predominantly female in both primary dentition (59.3%) and mixed dentition (78.6%), although these differences were also not statistically significant.

Table 9 presents the results of the presence and severity of malocclusions. Moderate degree was observed in two thirds of the males (67.0%) and approximately one third of the females (33.3%), (*p* = 0.787). No differences were detected regarding the presence of pain/discomfort or eating difficulties between males and females (*p* > 0.05).

## 4. Discussion

This work consists of an epidemiological study carried out in an immigrant population housed in the CETI of Melilla. Most of the refugees examined were of Syrian origin. The problem of migration of the Syrian people is of exceptional importance given the number of people who left this country. In 2015, approximately 4.6 million Syrian refugees are estimated to have fled to Turkey, Lebanon, Jordan and other countries. In 2018, more than 5.6 million Syrians fled the country as refugees, and 6.1 million are displaced within Syria, Half of the people affected by the terrible results of the war are children [31,32].

### 4.1. Caries

In the study group in Melilla, a high average number of decayed lesions was detected and more than half of the children in the mixed dentition required restorative treatment.

The study group reported a low frequency of toothbrushing, which causes plaque accumulation on tooth surfaces.

A study conducted in 2013 found a prevalence of dental caries in 113 children residing in Sao Tome, in the deciduous dentition of 58.9% and the mean dft 1.9 (±2.25) and in the permanent dentition was 38.8% and the mean DMFT was 0.9 (±1.55) [33].

Bourgeois and Llodra evaluated children from nine countries in four WHO regions (Cambodia, Greece, India, Indonesia, Kenya, Philippines, Morocco, Myanmar and Vietnam), observing a SiC index of 2.76 in a group of children aged 11–13 years and SiC index in children aged 6 years, ranging from 0.53 to 2.76, which is much lower than the value obtained in the present study [34].

Figure 1 shows the prevalence of caries in primary teeth in various countries. The results of the present study show higher values than those found in Portugal and Greece and lower than those identified in the Philippines.

Figure 2 presents the prevalence of caries in permanent teeth. The results of the present study in refugees in the CETI in Melilla show higher prevalence than in the groups studied in Portugal, but less high than observed in Myanmar. Comparisons between the studies should be made with caution given the differences in the type of sampling and some methodological aspects in which the studies differ. In a total sample of 2160 children aged 1–5 years, they observed a caries prevalence of 52.3% and 52.0% of children aged 11–13 years who had clinical signs of caries [34].

Branco analysed 263 pupils belonging to the Coimbra Oeste school cluster. Of the children observed, 135 were boys (51.3%) and 128 were girls (48.7%), aged between 6 and 11 years. Considering the 6064 teeth observed, 54.5% belonged to the deciduous dentition and 45.5% to the permanent dentition. In the deciduous dentition, 6.42% of the teeth were decayed, 1.18% were lost due to caries and 0.42% were restored. In the permanent dentition, 1.08% of the teeth were decayed, 0.07% were lost due to caries and no teeth were filled [35].

In the children examined in Melilla, it was observed that in the deciduous dentition (children under 6 years of age), 33.7% of the children had teeth with caries and the mean cod index was 6.3. In the mixed dentition, 51.0% of the children had caries and a mean cod index of 4.7 and a mean DMFS index of 2.0. In the permanent dentition, the percentage of decayed teeth was 15.9% with a mean DMFS index of 4.7. These values found were shown to be considerably higher than those of the previously analysed studies.

However, a 2016 study also carried out in the CETI in Melilla presented results of caries prevalence higher than ours in all the groups studied. In ages 5–7 and 8–10 years, caries prevalence was 75.0%, in children aged 11–13 years, it was 60.0% [30]. Further studies are required to identify the causes of the differences between the various studies conducted in Melilla, it is possible that given the mobility of the population staying in the CETI groups of different socio-economic and health conditions may be in this stay and may influence the results of studies conducted in different years.

### 4.2. Significant Caries Index

The Significant Caries Index (SiC) was proposed in 2000 to investigate the individuals at highest caries risk in each population. The SiC index is the average DMFT of one third of the study group with the highest caries prevalence [36]. In children aged 12 to 17 years housed in the CETI, we obtained a SiC value of 9.3 teeth, a SiC value of 10.1 in children aged 6 to 11 years and a SiC value of 6.4in children under 6 years. These results coincide with Brathal’s observation that a small part of the population concentrates a high percentage of caries experience. The development of preventive and care programmes for high-risk groups of children may be a useful strategy for the control of dental caries.

Another study, which analysed a Spanish and a migrant population, observed a SiC index of 2.8 in the Spanish population and 3.7 in the migrant population at 12 years of age and values of 4.4 in the Spanish population and 6.9 in the migrant population at 15 years of age, these values being much lower than those of our study [37].

### 4.3. Restauration Index

The Oral Health Survey in Spain 2010 showed that in the primary dentition, children aged 5–6 years have a low restorative attendance (RI 24.8%). At the age of 12 years, the DMFT with an RI of 52.7% and in the cohort of 15 years with an RI of 60.5%. When comparing the RI between Spanish and foreign children, the same author observed that in the age group 5–6 years (temporary), the RI was higher in Spanish children (27.5%) than in foreign children (18.8%), as well as in the age group 5–6 years (permanent) 18.7% in Spanish children and 7.0% in foreign children. The same was observed at ages 12 and 15 years, 55.7% and 63.7% in Spanish children, respectively, and 45.8% and 43.3% in foreign children, respectively [38]. In the present study, the restoration index presented low values at all ages (<3%), which is lower than that observed in other studies. This is consistent with the results of other studies showing a significant percentage of untreated teeth, particularly in population groups with limited resources and limited access to health services [39].

### 4.4. Treatment Needs

It is important to consider that children from low-income countries have a higher caries risk, quality of life problems and more dental treatment needs compared to children with a higher standard of living [39].

Llodra-Calvo, in the same previous study, showed that in Spanish and foreign children aged 5–6, 12 and 15 years, the greatest need for treatment is concentrated in restorations of a 154 surface [40].

In the present study, conducted in Melilla, in all ages, we observed a greater need for restorative treatment of three or more cavities. It is important to sensitise children, adolescents and parents to the importance of this, as childhood is a key period in the development of self-care, intersection and resolution of oral problems in a timely manner.

### 4.5. Periodontal Status

Periodontal diseases can affect children, adolescents and adults, with periodontitis resulting in the loss of bone and tooth-supporting tissue leading to discomfort, chewing difficulties and prosthetic problems [39]. Periodontal diseases are significantly affected by socioeconomic variables [41].

In a study conducted in Nigeria comparing the oral hygiene of two groups of children (those attending private and public schools), a total of 598 students were analysed, of which 300 (50.2%) were from public schools and 298 (49.8%) were from public schools, with the majority of participants (81.0%) between 11 and 15 years of age. It was concluded that a higher percentage of children from public schools had worse oral hygiene conditions (36.7%) compared to those from public schools (7.4%) and that access to oral hygiene care was reported to be better among children from public schools [42].

The Community Periodontal Index (CPI) recommended by the WHO for children over 12 years of age was used to record the periodontal status of the population of children at the CETI in Melilla, defined by sextants with healthy periodontal tissues, haemorrhagic or with calculus. The analysis of the CPI of the population examined at CETI showed a higher prevalence of haemorrhagic sextants (mean 3.9 and SD 2.5) with teeth 16, 26 and 46 being the most haemorrhagic.

The results of the present study showed that the mean number of decayed teeth aged 12–17 years was (DMFT 4.7 ± 4.2), not meeting the WHO oral health target for the year 2020 for 12 year olds.

In the group of children 6–11 years, the cod index was (4.7 ± 3.9), also not meeting the WHO oral health target for the year 2020 for 6 year olds.

Considering health not only as the absence of illness and knowing what the WHO proposes as the presence of wellbeing can be complemented with the idea that health allows the development of capacities, which includes the affective capacities of people and their link with the social group with which they interact. This concept of health requires recognition of fundamental collective rights. The refugee status of children limits their right to general health and oral health. The work carried out in the CETI of Melilla seeks to provide conditions that allow for.

### 4.6. Need for Resources and Specific Oral Health Program

With the data presented here, there is a clear need for oral treatments to be carried out in the affected population. With the current resources of the dental clinic located in the CETI, they can be carried out, but the needs are many. Strategies should be aimed at oral health programmes that place special emphasis on oral health promotion and prevention as indispensable elements. Such programmes were shown to have a positive impact on visits to the dentist as well as on changing brushing habits and oral health [43,44].

So far, we can only draw on volunteer dentists who work in the clinic on a completely altruistic basis. Given the high need for treatment, campaigns are carried out in which postgraduate students are involved, which complements them in both technical and human training. There are many economic resources necessary to have dentists on the payroll, which is why we are forced into this situation.

## 5. Conclusions

In conclusion, we could say that the prevalence of caries was high in the primary dentition with a relatively high average number of decayed teeth.

For the permanent dentition, the caries rate was also high with a significant need for restorative treatment, and so, it is necessary to work in the dental clinic of the CETI of Melilla in order to solve oral problems in people in a highly vulnerable situation such as refugee children. With these data, the Oral Health Objectives set by the WHO for the year 2020 were not met for these children.

The findings suggest that poor oral hygiene practices and limited access to restorative treatments contribute to the high prevalence of dental caries among this population. The comparative data from other studies highlight the importance of a global approach to addressing oral health issues among children and adolescents.

This is why oral health education programmes would be extremely useful in Melilla, considering the state of the oral cavity of the people housed in the CETI.

## Figures and Tables

**Figure 1 children-10-00888-f001:**
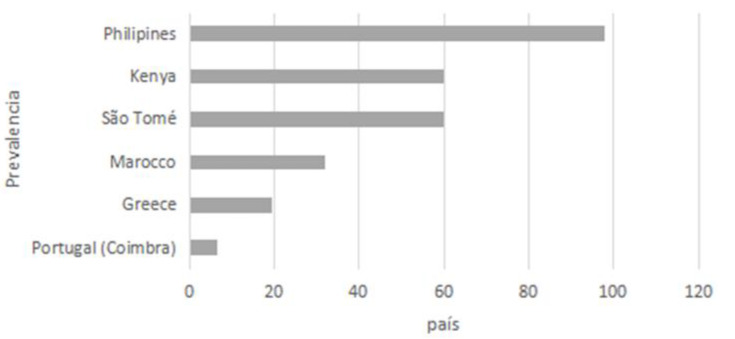
Caries Prevalence in primary dentition in several countries [34].

**Figure 2 children-10-00888-f002:**
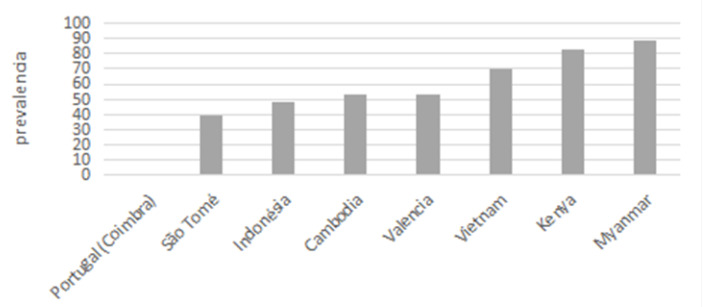
Caries Prevalence in permanent dentition in several countries.

**Table 1 children-10-00888-t001:** Sociodemographic description (*n* = 198).

COUNTRY	*n*	%
Syria	172	86.9
Palestine	6	3.0
Morocco	12	6.1
Ivory Coast	2	1.0
Liberia	2	1.0
Algeria	4	2.0
**SEX**		
Boys	114	57.6
Girls	84	42.4
**AGE (YEARS)**		
1–3	37	18.7
4–5	31	15.7
6–8	51	25.8
9–11	38	19.2
12–13	21	10.6
14–17	20	10.1
**TYPE OF DENTITION**		
Temporal	68	34.3
Mixed	89	44.9
Permanent	41	20.7

**Table 2 children-10-00888-t002:** Clinical description of the dental status of the study group (*n* = 198) according to dental age groups (<6 years; 6–11 years; 12–17 years).

	<6 Years (n = 68; 34.3%)		6–11 Years (n = 89; 44.9%)		12–17 Years (n = 41; 20.7%)	
Temporary Teeth	Mean	SD	Mean	SD	Mean	SD
decayed	6.3	6.3	4.7	3.9	0.0	0.0
filled	0.0	0.0	0.0	0.0	0.0	0.0
missing	0.1	0.5	0.8	1.3	0.0	0.0
dmft	6.4	6.3	4.7	3.9	0.0	0.0
Temporary-RI * %	0.0	0.0	0.0	0.0	0.0	0.0
Permanent teeth						
Decayed	0.1	0.6	2.0	2.0	4.4	4.0
Filled	0.0	0.0	0.0	0.2	0.2	1.1
Missing	0.0	0.0	0.0	0.0	0.1	0.3
DMFT	0.1	0.6	2.0	2.0	4.7	4.2
Permanent-RI * %	0.0	0.0	2.0	0	2.9	13.0
RI *-Total %	0.0	0.0	0.5	3.5	2.8	13.0
SiC **	6.4	6.4	10.1	6.0	9.3	2.6

* RI: Restoration Index: ratio of the number of filled teeth in the caries indices. ** SiC: Mean of the third of the group with the highest DMFT.

**Table 3 children-10-00888-t003:** Description of dental treatment needs of the study group (*n* = 198) by dental age groups (<6 years; 6–11 years; 12–17 years).

	<6 Years Temporal Dentition (n = 68; 34.3%)		6–11 Years Mixed Dentition (n = 89; 44.9%)		12–17 Years Permanent Dentition (n = 41; 20.7%)	
	Mean	SD	Mean	SD	Mean	SD
Fillings	4.2	4.5	4.8	3.0	4.1	3.7
Invasive Treatment	2.3	4.5	1.8	3.0	0.3	0.9
Space Mantainer	0.0	0.1	0.3	1.0	0.0	0.0
	N	%	N	%	n	%
Distribution of need of fillings						
None	24	35.3	10	11.2	11	26.8
1–2 fillings	6	8.8	12	13.5	5	12.2
3 or more	38	55.9	67	75.3	25	61.0
Distribution of need for invasive treatment (endo or exodontics)						
None	42	61.8	38	42.7	34	82.9
1–2 treatments	5	7.4	27	30.3	5	12.2
3 or more	21	30.9	24	27.0	2	4.9

**Table 4 children-10-00888-t004:** Clinical description of periodontal status in subjects aged 12–17 years (*n* = 41).

Mean of Sextants and Value	Mean	SD
Number of sextants with CPI = 0	2.0	2.6
Number of sextants with CPI = 1	3.9	2.5
Number of sextants with CPI = 2	0.1	0.3
Needs of periodontal treatment	n	%
None	11	26.8
Prophylaxis and Oral hygiene	30	73.2

**Table 5 children-10-00888-t005:** Dental treatment needs in study group (*n* = 198) in the different dental age groups (<6 years; 6–11 years; 12–17 years) by sex.

	<6 Years Temporal Dentition (*n* = 68; 34.3%)				6–11 Years Mixed Dentition (*n* = 89; 44.9%)				12–17 Years Permanent Dentition (*n* = 41; 20.7%)			
Needs of	Boys (*n* = 41)		Girls (*n* = 27)		Boys (*n* = 47)		Girls (*n* = 42)		Boys (*n* = 26)		Girls (*n* = 15)	
	Mean	SD	Mean	SD	Mean	SD	Mean	SD	Mean	SD	Mean	SD
Fillings	4.2	4.7	4.2	4.4	4.5	3.0	5.1	3.1	4.1	3.7	4.1	3.7
Crowns	0.1	0.3	0.0	0.0	0.0	0.0	0.0	0.2	0.0	0.0	0.0	0.0
Total restorations	4.3	4.7	4.2	4.4	4.5	3.0	5.2	3.1	4.1	3.7	4.1	3.7
Invasive Treatments	3.0 *	4.7	1.2 *	2.4	2.1	2.7	1.5	2.2	0.3	0.8	0.3	0.6
Space Mantainers	0.0	0.2	0.0	0.0	0.5 *	1.3	0.1 *	0.5	0.0	0.0	0.0	0.0

* Statistically significant differences between men and women of the same age group after Student’s test analysis (*p* < 0.05).

**Table 6 children-10-00888-t006:** Comparison of Periodontal status according to the average Community Periodontal Index (CPI) in the age group 12–17 years (*n =* 41).

	Boys (*n* = 26)		Girls (*n* = 15)	
	Mean	SD	Mean	SD
Healthy CPI = 0	2.2	2.5	1.7	2.7
Bleeding gums CPI = 1	3.9	2.5	4.1	2.7
Calculus CPI = 2	0.0	0.0	0.1	0.5

**Table 7 children-10-00888-t007:** Comparison of dental visits and toothbrushing by sex in those aged 12 years and over (*n* = 41).

	Boys (*n* = 26)		Girls (*n* = 15)	
Visits to the Dentist	*n*	%	*n*	%
No in last year	5	71.4	2	28.6
Sometimes in last year	9	64.3	5	35.7
Never	12	60.0	8	40.0
Toothbrushing	n	%	n	%
Rarely	11	64.7	6	35.3
Every week	5	83.3	1	16.7
Daily	10	55.6	8	44.4

**Table 8 children-10-00888-t008:** Comparison of the dental treatment needs of the study sample (*n* = 198) according to sex in the different age groups (under 6 years; 6–11 years and 12–17 years).

	<6 Years Temporal Dentition (*n* = 68; 34.3%)				6–11 Years Mixed Dentition (*n* = 89; 44.9%)				12–17 Years Permanent Dentition (*n* = 41; 20.7%)			
Needs of Fillings	Boys		Girls		Boys		Girls		Boys		Girls	
	*n*	%	*n*	%	*n*	%	*n*	%	*n*	%	*n*	%
None	16	39.0	8	29.6	6	12.8	4	9.5	7	26.9	4	26.7
1–2	3	7.3	3	11.1	7	14.9	5	11.9	3	11.5	2	13.3
3	22	53.7	16	59.3	34	72.3	33	78.6	16	61.5	9	60.0

**Table 9 children-10-00888-t009:** Comparison of malocclusion type, pain and eating problems by sex in the permanent dentition 12–17 years (*n* = 41).

Type of Malocclusion	Boys		Girls		Total
	*n*	%	*n*	%	*n*
None	1	100.0	0	0.0	1
Slight	10	59.0	7	41.2	17
Moderate	14	67.0	7	33.3	21
Severe	1	50.0	1	50.0	2
Total	26	63.4	15	37.0	41
Pain/discomfort	Boys		Girls		Total
	*n*	%	*n*	%	*n*
Never	26	65.0	14	35.0	40
Sometimes	0	0.0	1	100.0	1
Total	26	63.4	15	36.6	41
Problems at eating	Boys		Girls		Total
	*n*	%	*n*	%	*n*
Never	26	65.0	14	35.0	40
Hardly ever	0	0.0	1	100.0	1
Total	26	63.4	15	36.6	41

## Data Availability

Data are available by corresponding authors.

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
