# Peer review of "Oral Health in migrants children in Melilla, Spain"

_children, 2023, doi:10.3390/children10050888_

Round 1
Reviewer 1 Report
This study has some potentially interesting findings. However, there are certain areas that require more attention. My comments are listed below:
1. How did authors estimate the sample size?
2. How long were participants housed in the CETI of Melilla?
3. What is the meal menu for children housed in the CETI of Melilla?
4. Are children housed in the CETI of Melilla provided with toothbrushes? Furthermore, who usually brushes the teeth for infants?
5. What percentage of participants had systemic diseases or poor health conditions?
6. Were calibration of examiners and reproducibility test performed? 
7. Authors should add the descriptions of the procedure and condition of oral examination in Methods section.
8. In Discussion section, authors described that “A study conducted in 2013, found a prevalence of dental caries in 113 children residing in Sao Tome, in the deciduous dentition of 58.9% and the mean CPOD 1.9 (± 2.25) and in the permanent dentition was 38.8% and the mean CPOD was 0.9 (± 1.55) [33].” Please provide a brief description of CPOD. 
9. In conclusion section, authors insisted that it is necessary to work in the dental clinic of the CETI of Melilla in order to solve oral problems in people in a highly vulnerable situation such as refugee children. Do the authors have strategies on how to increase the number of dentists working in the CETI of Melilla? Please add the discussion of them.
Author Response
Please, see the attachment.

Reviewer 2 Report
1. The introduction raises important concerns about the precarious state of oral health among immigrant children and emphasizes the need for an oral health program with a marked social character. However, there are a few areas that could be critiqued.
Firstly, the paragraph lacks specific information about the oral health status of immigrant children in Melilla. The author assumes that their oral health is precarious, but no objective data is presented to support this claim.
Secondly, the paragraph provides no details about the proposed oral health program. It is unclear what specific interventions the program will include or how it will be implemented.
Lastly, the paragraph could benefit from more information about the resources required for the program. While the author mentions the need to quantify resources, no information is provided about what those resources might be or how they will be obtained.
2. The method presents the type of study and design used to assess the oral health status of children and adolescents living in refugee conditions in the CETI of Melilla, Spain. However, there are a few areas that could be critiqued.
Firstly, the paragraph does not provide any information about the sample size or the sampling method used in the study. This information is crucial to determine the generalizability of the study results.
Secondly, the paragraph does not provide any details about the ethical considerations taken into account during the study. For example, there is no information about obtaining informed consent from the participants or their guardians.
Thirdly, the paragraph provides a brief description of the statistical methods used in the study but lacks details about the analytical techniques used to examine the data. It is unclear how the data was analyzed to arrive at the study findings.
3. The discussion highlights the alarming prevalence of dental caries among children and adolescents in the study group in Melilla, Spain. The study found a high average number of cavitated lesions, and more than half of the children in the mixed dentition required restorative treatment. The study group also reported a low frequency of toothbrushing, leading to plaque accumulation on tooth surfaces.
Comparative data from other studies are presented to contextualize the findings. A study conducted in 2013 in Sao Tome found a prevalence of dental caries in the deciduous dentition of 58.9% and a mean CPOD of 1.9. In the permanent dentition, the prevalence was 38.8%, and the mean CPOD was 0.9. In contrast, Bourgeois and Llodra evaluated children from nine countries in four WHO regions, observing a SiC index of 2.76 in a group of children aged 11-13 years and a SiC index ranging from 0.53 to 2.76 in children aged six years. The SiC index values reported in these studies are much lower than those obtained in the present study.
Overall, the paragraph underscores the urgent need for interventions to improve the oral health status of children and adolescents in Melilla. The findings suggest that poor oral hygiene practices and limited access to restorative treatments contribute to the high prevalence of dental caries among this population. The comparative data from other studies highlight the importance of a global approach to addressing oral health issues among children and adolescents.
Author Response
Please, see the attachment.

Round 2
Reviewer 1 Report
The comments were satisfactorily addressed, and I anticipate an acceptance of the revised manuscript.